# University Students Attitudes toward Same-Sex Marriage Adoption in Taiwan

**Hui-Chi Wang * and Hua-Chang Lee ***

Faculty of Department of Social Work, Chaoyang University of Technology, Taichung 41349, Taiwan
* Correspondence: whc@cyut.edu.tw (H.-C.W.); hclee@cyut.edu.tw (H.-C.L.)

**Abstract:** Taiwan is the first country in Asia to recognize the legal rights of same-sex couples to get married. Although same-sex marriage has been legal in Taiwan since May 2019, the same-sex marriage family was not allowed to adopt child legally; only stepchild adoption was permitted. This is still a very controversial issue, so this study intended to understand the views of Taiwanese college students, whose voices should be heard and whose opinions should be valued by legislators. To investigate this issue, a questionnaire was constructed, and 440 objects were collected. The questionnaire regarding attitudes toward same-sex marriage adoption consisted of three dimensions: "Worry and against", "Idea Recognition" and "Action Support". Each dimension has good reliability. The internal consistent coefficients (Cronbach's $\alpha$) were 0.86, 0.93, and 0.94. The responses reveal that college students in Taiwan have a relatively positive attitude towards same-sex marriage adoption, and college students who are biologically female, non-heterosexual, non-Christian, major in social work, and are acquainted with the LGBT community have more positive attitudes. College students' same-sex marriage and adoption attitudes can be predicted by biological sex, sexual orientation, religion, grade, whether they major in social work, have contact experience with the LGBT community, contact experience with adoption, and same-sex parenting concepts. The same-sex parenting concept is the most important predictor variable, which means that the judgment about whether same-sex marriage couples can bear the responsibility of raising children is the most critical factor affecting the attitude of same-sex marriage adoption.

**Keywords:** attitude; experience of contacting LGBT; same-sex parenting; same-sex marriage adoption

## 1. Introduction

There are 33 countries or regions in the world that legally allow same-sex marriages. Most which allow same-sex marriage also allowed same-sex adoption either immediately or subsequently, while a few only allow stepparent adoption or have not yet legalized same-sex adoption. Sixteen out of thirty European countries (including Iceland, France, Spain) have both legalized same-sex marriage and same-sex adoption rights. Another seven countries have recognized registered same-sex relationship, but only two of them (Estonia and Slovenia) allow same-sex adoption. The remaining seven governments (such as Bulgaria, Poland) did not pass same-sex marriage or same-sex adoption laws (European Social Survey 2022).

It seems that the right to adoption by same-sex families would be gradually granted after the legalization of same-sex marriage, but Taiwan, since 2019, has not. It causes the problems such as some same-sex couples being afraid of getting married because they were not allowed to adopt a child legally. Even if they could adopt a child legally before marriage, the parental rights over the child were limited to only one of them. Children are at risk when one of their parents fails to have a legal relationship with them (Scherman et al. 2020). They could miss out on inheritance rights, retirement benefits, health insurance, and even lose the relationship with the non-legal parent if their same-sex parents end the marriage (Goldberg and Kuvalanka 2012; Acosta 2017; Mason 2018).

Could we rely on young people in Taiwan to drive the trend forward? Have the law makers who passed the same-sex marriage laws satisfied the young people of Taiwan's positive perceptions of same-sex adoption? What are the attitudes of social work students, i.e., those who will become adoption professionals, towards same-sex adoption? These are the primary questions that sparked this study.

### 1.1. Legislative Background of Same-Sex Marriage in Taiwan and Restrictions on Same-Sex Marriage Adoption Rights

Movements related to same-sex marriage in Taiwan began when Chi Chia-wei proposed a civil marriage with his same-sex partner in 1986 but was rejected. This was the beginning of his involvement in the gay rights movement (Liu 2017). It also started a series of petitions and protests for the same-sex marriage legislation in Taiwan.

Taiwan's marriage norms are derived from the Civil Code of Taiwan, which does not provide legal recognition for same-sex marriage. To legalize same-sex marriage, the Taiwan Partners' Rights Promotion Alliance has actively pushed for the "Draft Legislation on Diversified Families" since 2012. This legislation includes three sets of bills aimed at amending civil law to allow same-sex couples to marry equally and obtain legal protection for non-married families. The Marriage Equality Draft, proposed by Li-Chiun Cheng, Mei-Nu Yu, Bi-khim Hsiao, and other legislators, advocates for marriage to be independent of gender and for the non-discrimination of multi-gender adopters based on the best interests of the child. This draft was jointly signed by 22 members of the Taiwan Legislative Yuan on 23 October 2013.

It passed the first reading, but at that time, the legalization of same-sex marriage and the rights of adopted children in particular attracted the most public attention and comments and were the most controversial. In 2015, Chi Chia-wei and the Taipei City government submitted an application for constitutional interpretation to the judge. The judge of the Republic of China Judiciary compiled the marriage seal for the relatives in the "Civil Law", a permanent bond of intimacy and exclusivity, considered whether it violates Article 22 of the Constitution of the Republic of China to guarantee the freedom of marriage and Article 7 to guarantee "the right to equality", and then declared, in the 2017 Interpretation, "no". Within 2 years from the date of the announcement of the interpretation, the amendment or enactment of laws related to same-sex marriage were completed in accordance with the intention of the interpretation (Constitutional Court 2017). Due to the fierce dispute between Taiwan's opposition and support for same-sex marriage, in the national referendum held by the Central Election Commission in 2018, three cases were related to same-sex marriage, including Case 10, "Do you agree that civil law marriage should be limited to the union of a man and a woman?"; Case 12, "Do you agree to protect the rights and interests of two people of the same sex to live together permanently in other forms than the marriage provisions of the civil law?"; and Case No. 13, "Do you agree that the civil law guarantees same-sex marriage rights?" According to the results of the referendum (Central Election Commission 2018), the agree rate of case 10, 12, and 13 were 72.48% (pass), 61.12% (pass), and 32.74% (fail). This means that people in Taiwan tend to keep the man–woman marriage definition in civil law (case 10); same-sex marriage rights should not be guaranteed by civil law (case 13); it could be recognized in other forms than the civil law (case 12). The Act for Implementation of J.Y., Interpretation No. 748, which recognized the rights of same-sex marriage, came into force on 4 May 2019. To avoid further confrontation between the "pro/con" sides, Act 748 limits the right of same-sex marriage spouses to jointly adopt unrelated children; that is, they cannot jointly adopt children like monogamous marriages; only one party in a same-sex marriage can adopt the other's biological children.

### 1.2. Perceptions of Parenting in Same-Sex Parents

Chiang and Su (2022) found that the issue of same-sex marriage adoption remains highly controversial in Taiwan, with opponents using "family destruction metaphors" to

reinforce negative attitudes towards same-sex marriage and same-sex families. Some argue that same-sex parenting deviates from the traditional heterosexual couple model, leading to concerns about a lack of role models, bullying, or LGBTQ identities. However, studies show that sexual orientation is not a critical factor for good parenting, and that parental relationship and family stability are more important for a child's growth. A meta-analysis by Suárez et al. (2022) even found that same-sex parents can have a positive influence on children's development and be competent caregivers.

### 1.3. Factors Affecting Perceptions of Same-Sex Parenting

#### 1.3.1. Gender

Many studies found that gender is an important factor for same-sex parenting. Men express negative attitudes towards same-sex marriage and same-sex parenting more than women (Costa et al. 2014a, 2014c; Massey et al. 2013). Gender would have more impact to same-sex adoption attitudes in those countries with less progressive legislation. Higher educated women and younger man are more willing to accept same-sex family and same-sex adoption (Webb et al. 2017).

#### 1.3.2. Religious Beliefs

Religious beliefs were found one of the strongest predictors of negative attitude to same-sex adoption. Some Christian fundamentalist and Catholic institutions refuse to work with same-sex applicants, while Jewish-affiliated and Lutheran organizations are more open (Costa et al. 2014b; Scherman et al. 2020).

In Taiwan, opposition to same-sex marriage was led by the "Taiwan Religious Group Love Family Alliance" (also known as the Family Protection League), a Christian-led organization that was established in 2015. The Faith and Hope Alliance, a political party with a Christian foundation, also launched a referendum opposing same-sex marriage under the guise of protecting families and children (Wang 2018). However, there are also many Christians in Taiwan who support same-sex marriage, such as the "Tongguang Gay Church", Taiwan's first gay Christian church, and Pastor Chen Si-hao of the Presbyterian Church. Additionally, both the pro and con sides have garnered support from some Buddhist masters. It is clear that the religious community in Taiwan holds diverse views on the issue of same-sex marriage and same-sex adoption.

#### 1.3.3. Gender Role

Men often conform to traditional gender roles in order to maintain their higher social status and roles within patriarchal social structures (Korpi et al. 2013). This can explain why younger men are less affected by traditional gender norms, as they are more likely to be exposed to different family structures (Gilbert 2014), which makes them more open to the idea of same-sex parenting. Lesbian parenting is more acceptable due to female stereotypes and trust in their innate parenting abilities (Imaz 2017).

#### 1.3.4. Contact Experiences and Legislation

The frequency and direction of contact will affect attitudes towards same-sex marriage and same-sex adoption (Webb and Chonody 2014). In addition, exposure to the emotional consequences of unequal rights will also increase support for same-sex marriage (Case and Stewart 2010). Progressive government legislation could also affect views on same-sex parenting. According to Van de Rozenberg and Scheepers (2022), only countries with more progressive legislation on same-sex relationships and highly educated individuals do not deny same-sex adoption rights.

### 1.4. Professional Barriers to Same-Sex Adoption

McCutcheon and Morrison (2014) introduce the concept of "homonegativity", which refers to stereotypes and negative attitudes towards gays and lesbians from social workers and other adoption professionals. Negative attitudes of social workers will make

them favor heterosexual couples and lead to different treatment of gay adoptees. These biases may manifest in subtle ways or overt discrimination due to institutional factors (Scherman et al. 2020).

In conclusion, social workers may favor heterosexual couple applicants and ignore gay adoption applicants. Some adoption social workers may also neglect to mention the waiting time or advantages of gay adopters when contacting adopters.

### 1.5. Purposes

This study aims to understand the attitudes of college students in Taiwan towards same-sex adoption. The voices of young people should be heard; they have a great impact on future legislation and policy. In addition, this study investigated the influence of college students' biological sex, sexual orientation, religious belief, grade, major in social work, experience with gays, experiences with adoption, and same-sex parenting attitudes on attitudes toward same-sex marriage and adoption.

### 1.6. Theoratical Thinking

Breckler (1984) proposed, through empirical research, that attitudes consist of three core elements: "Affection," "Behavior," and "Cognition." This provided the widely accepted ABC model of attitudes today. Attitudes towards same-sex adoption in this study will also be developed as tools based on Breckler's theory and will be used to validate his theory.

## 2. Materials and Methods

### 2.1. Participants

Table 1 shows the final sample consisted of 440 (95 physical males and 345 physical female) students—1st grade, 76 (17.3%); 2nd grade, 83 (18.9%); 3rd grade, 186 (42.3%); 4th grade, 69 (15.7%); and graduate school, 25 (5.7%).

**Table 1.** Sociodemographic characteristics of participants.

| Variables | Levels | Counts | % | Variables | Level | Counts | % |
|---|---|---|---|---|---|---|---|
| sex N = 440 | Male | 95 | 21.6 | Sexual tendency N = 435 | hetero | 334 | 76.8 |
| | Female | 345 | 78.4 | | homo | 16 | 3.7 |
| religious | I-Kuan Tao | 9 | 2.1 | | bisexual | 85 | 19.5 |
| N = 432 | Buddhism | 51 | 11.8 | grade | 1 | 76 | 17.3 |
| | Christians | 39 | 9.0 | N = 439 | 2 | 83 | 18.9 |
| | None | 234 | 54.2 | | 3 | 186 | 42.4 |
| | Taoism | 99 | 22.9 | | 4 | 69 | 15.7 |
| Gay | No contact | 47 | 10.7 | | graduates | 25 | 5.7 |
| contact | Not familiar | 115 | 26.3 | Major in social | No | 271 | 61.7 |
| N = 438 | Familiar | 276 | 63.0 | work | related | 19 | 4.3 |
| Adoption | No contact | 354 | 80.6 | N = 439 | yes | 149 | 33.9 |
| contact | not familiar | 47 | 10.7 | | | | |
| N = 439 | Familiar | 38 | 8.7 | | | | |

Most (N = 334, 75.9%) of the participants identify as heterosexual, with 3.6% (N = 16) identifying as homosexual and 19.3% (N = 85) identifying as bisexual.

Those major in social work mounted to 33.9% (N = 149), and other helping professions, 4.3% (N = 19); most of the participants (N = 271, 61.7%) are not majoring in social work. Experience of gay contact was 10.7% (N = 47) having no contact; while 115 participants (26.3%) were not familiar; and rest of the participants (N = 276, 63%) were familiar with gay or LGBT ones.

The religious beliefs of the samples include 9 Yiguan Taoists; 51 Buddhists; 99 Taoists; 39 Christians; and 234 of no specific religious beliefs.

### 2.2. Procedure

A total of 440 college students were invited to fill out the "Same-sex Marriage Adoption Attitude Scale", constructed by the researchers on the SurveyCake online platform from September to October 2022. Filling out the scale is voluntary and anonymous, and participants were asked to answer 8 demographic information questions, 3 questions about same-sex parenting, and 16 questions about same-sex marriage adoption. The responses were recorded online and analyzed after the end of the survey. The abstract would be sent to the ones who were interested in this investigation and gave their contact information.

### 2.3. Measures

Two Likert type 5-point scales were constructed in this study. The first one is the Same-sex parenting Concept Scale, which consists of 3 items, and the internal consistency (Cronbach $\alpha$) is 0.89; the other is the Same-sex Marriage Adoption Attitude Scale, which has 16 items, which were written according to the Breckler's theory of attitude (Breckler 1984).

The Same-sex Marriage Adoption Attitude Scale was divided into 3 subscales after the procedure of factor analysis, namely, "Worry/Against" (4 items), "Action Support" (7 items), and "Idea Recognition" (5 items); the internal consistencies (Cronbach $\alpha$) were 0.86, 0.93, and 0.94, in sequence. All the subscales have very good internal consistency reliabilities. All items are rated on a 5-point Likert Scale (1 = strongly disagree, 5 = strongly agree). The factor analysis procedures are present as Table 2.

**Table 2.** Factor analysis procedure and item means.

| | (N = 435) | Scores | | Factor Loading | | |
|---|---|---|---|---|---|---|
| **Item** | **Content** | **Mean** | **SD** | **F1 [a]** | **F2** | **F3** |
| 1 | Discriminated against | 2.05 | 0.983 | [#] | −0.35 | −0.73 |
| 2 | Worry about be rejected | 2.39 | 1.070 | | −0.26 | −0.60 |
| 3 | Worry about other people knowing | 2.01 | 0.981 | | −0.21 | −0.73 |
| 4 | Worry about violating moral | 1.98 | 1.015 | −0.22 | −0.23 | −0.76 |
| 5 | Eliminate social prejudice | 4.35 | 0.757 | 0.46 | 0.39 | 0.40 |
| 6 | Respect for Gay Families Adoption | 4.41 | 0.742 | 0.52 | 0.50 | 0.44 |
| 7 | College should advocate | 4.26 | 0.832 | 0.68 | 0.38 | 0.31 |
| 8 | College set related classes | 3.99 | 0.890 | 0.77 | | |
| 9 | I am willing to help | 3.89 | 0.930 | 0.81 | 0.25 | |
| 10 | I will encourage them | 4.13 | 0.825 | 0.67 | 0.46 | 0.27 |
| 11 | Willing to help adoption | 3.92 | 0.919 | 0.76 | 0.31 | |
| 12 | Gays have the right | 4.34 | 0.848 | 0.34 | 0.81 | 0.34 |
| 13 | Lesbians have the right | 4.35 | 0.825 | 0.35 | 0.81 | 0.33 |
| 14 | Any sexual tendency has the right | 4.40 | 0.870 | 0.28 | 0.72 | 0.36 |
| 15 | Same-sex parents are a complete family | 4.37 | 0.850 | 0.41 | 0.61 | 0.38 |
| 16 | Love and support are healthy enough | 4.45 | 0.782 | 0.39 | 0.64 | 0.35 |
| | Eigenvalues after rotation | | | 4.00 | 3.84 | 3.21 |
| | Percentage explained | | | 25.0 | 24.0 | 20.1 |

[a] F1 = Action support; F2 = Idea Recognition; F3 = Worry/Against. [#] For clarity of reading, factor loadings below 0.2 were omitted.

The KMO value of the 16 topics this time is 0.92, which is very suitable for factor analysis. Using the principal axis factor method to extract three factors, the eigenvalues after rotation are 4.0, 3.84, and 3.21, each explaining 25%, 24%, and 20% of the original total variation, and the cumulative explanation is 69%. After examining the contents of the three factors, they are named as follows: F1 = Action support; F2 = Idea Recognition; F3 = Worry/Against. Although the three factors are all related to same-sex adoption attitudes, they show meaningful differences. Therefore, this study sums up the three

subscales separately, discusses them separately, and does not calculate the overall score of gay adoption attitudes. For the convenience of conceptual comparison, the three total scores are divided by the number of items so that the scores can be considered with the characteristics of a five-point scale.

## 3. Results

Out of the 440 college students surveyed on their attitudes towards same-sex marriage adoption (Table 2), the highest attitude score was for "Idea Recognition" (ranging from 4.34 to 4.45), followed by "Action Support" (ranging from 3.89 to 4.41), and finally "Worry/Against" (ranging from 1.98 to 2.39). Overall, the positive attitudes on the five-point scale were above the midpoint, while negative attitudes were below it. This suggests that college students tend to support same-sex marriage adoption through recognition and action, rather than opposition.

Table 3 shows that males score higher than females on the Worry/Against dimension but score lower than females on both the Idea Recognition and Action Support dimensions.

**Table 3.** *t*-test of college students' biological sex and same-sex marriage adoption attitude.

| Sex<br>Same-Sex Marriage Adoption | Male (N = 95) | | Female (N = 343) | | *t* | df | Sig |
|---|---|---|---|---|---|---|---|
| | Mean | SD | Mean | SD | | | |
| Worry/Against | 2.475 | 0.99 | 1.93 | 0.749 | 5.79 * | 436 | 0.000 |
| Action Support | 3.911 | 0.75 | 4.21 | 0.674 | −3.67 * | 436 | 0.000 |
| Idea Recognition | 4.185 | 0.86 | 4.44 | 0.707 | −2.99 * | 437 | 0.003 |

* $p < 0.05$.

On the Worry/Against dimension, the average value for heterosexuality is greater than that of homosexuality and bisexuality.

The average values for heterosexuals on the Action Support and Idea Recognition dimensions are significantly lower than those for homosexuals and bisexuals (Table 4). This suggests that heterosexuals tend to show more opposition and concern towards gay adoption attitudes, as well as less recognition of the idea and less support for action. However, even for the group of heterosexuals with the highest average, the average score for worry and opposition is only 2.17, which is still below the midpoint of the five-point scale. Overall, this indicates that college students' attitudes towards gay adoption are not predominantly negative; rather, heterosexual college students tend to have more worries and opposition than homosexuals and bisexuals.

**Table 4.** F-test of college students' personal sexual orientation and same-sex marriage adoption attitude.

| Sexual Orientation | | 1. Heterosexuality | 2. Homosexuality | 3. Bisexual | F | Sig | Posteriori Test |
|---|---|---|---|---|---|---|---|
| Same-Sex Marriage Adoption | | (N = 334) | (N = 16) | (N = 83) | | | |
| Worry/Against | Mean<br>SD | 2.17<br>0.86 | 1.55<br>0.65 | 1.66<br>0.60 | 16.125 * | 0.000 | 1 > 2, 3 |
| Action Support | Mean<br>SD | 4.05<br>0.71 | 4.50<br>0.62 | 4.44<br>0.50 | 13.851 * | 0.000 | 1 < 2, 3 |
| Idea Recognition | Mean<br>SD | 4.30<br>0.79 | 4.86<br>0.30 | 4.66<br>0.47 | 12.216 * | 0.000 | 1 < 2, 3 |

* $p < 0.05$.

Table 5 shows that participants who are more familiar with homosexuals exhibit higher levels of Idea Recognition and Action Support, as well as lower levels of Worry/Against attitudes.

**Table 5.** F-test of college students' gay contact experience and same-sex marriage and adoption attitudes.

| Homosexuality Contact Experience | | 1. Never Contacted | 2. Contact But Not Familiar | 3. Familiar | F | Sig | Posteriori Test |
|---|---|---|---|---|---|---|---|
| Same-Sex Marriage Adoption | | (N = 47) | (N = 115) | (N = 274) | | | |
| Worry/Against | Mean | 2.60 | 2.19 | 1.90 | 17.325 * | 0.000 | 1 > 2 > 3 |
| | SD | 0.87 | 0.86 | 0.77 | | | |
| Action Support | Mean | 3.88 | 3.98 | 4.25 | 10.218 * | 0.000 | 1, 2 < 3 |
| | SD | 0.80 | 0.75 | 0.63 | | | |
| Idea Recognition | Mean | 4.13 | 4.20 | 4.51 | 10.655 * | 0.000 | 1 < 2 < 3 |
| | SD | 0.88 | 0.87 | 0.64 | | | |

* $p < 0.05$.

According to Table 6, we can see that social work students score lower than non-social work students on the Worry/Against dimension and score higher on Action Support. However, there is no difference between the three groups in terms of Idea Recognition, with both groups scoring very high.

**Table 6.** F-test for social work students and same-sex marriage and adoption attitudes.

| Major in Social Work Same-Sex Marriage Adoption | | 1. No (N = 270) | 2. Related (N = 19) | 3. Yes (N = 148) | F | Sig | Posteriori Test |
|---|---|---|---|---|---|---|---|
| Worry/Against | Mean | 2.15 | 2.37 | 1.83 | 8.52 * | 0.000 | 1, 2 > 3 |
| | SD | 0.85 | 1.10 | 0.74 | | | |
| Action Support | Mean | 4.03 | 4.22 | 4.33 | 8.992 * | 0.000 | 1 < 2, 3 |
| | SD | 0.69 | 0.86 | 0.66 | | | |
| Idea Recognition | Mean | 4.33 | 4.27 | 4.50 | 2.471 | 0.086 | none |
| | SD | 0.77 | 0.87 | 0.69 | | | |

* $p < 0.05$.

Table 7 shows that all religions have low levels of Worry/Against, with no significant differences between them. Due to the large differences in sample size, no statistically significant differences were found in Action Support. However, in terms of Idea Recognition, Christian participants scored significantly lower than non-religious and Taoist participants.

**Table 7.** F-test of college students' religious beliefs and same-sex marriage and adoption attitude.

| Religious Same-Sex Marriage Adoption | | I-Kuan Tao (N = 9) | Buddhism (N = 51) | Christian (N = 39) | Non-Religious (N = 233) | Taoism (N = 99) | F | Sig | Posteriori Test |
|---|---|---|---|---|---|---|---|---|---|
| Worry/Against | Mean | 2.25 | 2.02 | 2.41 | 1.99 | 2.04 | 2.293 | 0.059 | |
| | SD | 0.81 | 0.72 | 1.08 | 0.81 | 0.84 | | | |
| Action Support | Mean | 3.65 | 4.07 | 3.99 | 4.19 | 4.21 | 2.284 | 0.060 | Christian < non-religious, Taoism |
| | SD | 0.83 | 0.66 | 0.92 | 0.66 | 0.65 | | | |
| Idea Recognition | Mean | 3.93 | 4.40 | 4.07 | 4.44 | 4.46 | 3.195 * | 0.013 | |
| | SD | 0.73 | 0.60 | 1.08 | 0.70 | 0.71 | | | |

* $p < 0.05$.

*Prediction on Adoption Attitudes*

In order to explore the predictive power of various variables on the three subscales of adoption attitudes, this study divides the predictive variables into three blocks. The first block includes biological sex, sexual orientation, and grade as demographic variables; the second block concerns contact experience and includes majoring in social work, gay contact experience, and adoption contact experience; the third block is about attitudes towards gay parenting. In this study, the predictive variables of the three blocks were sequentially

added into the regression model, and the changes in their predictive power were examined. Because the three subscales of Worry/Against, Action support, and Idea recognition show different adoption attitudes, they were predicted separately, and showed in Tables 8–10.

**Table 8.** Hieratical regression to Worry/Against by three blocks of predictors.

|  | Block 1: Demographic | | | Block 2: Experience | | | Block 3: Same-Sex Parenting | | |
|---|---|---|---|---|---|---|---|---|---|
|  | Beta | $t$ | $p$ | Beta | $t$ | $p$ | Beta | $t$ | $p$ |
| Biological sex | −0.204 | −4.355 * | 0.000 | −0.18 | −3.92 * | 0.00 | 0.00 | 0.03 | 0.98 |
| Sexual orientation | −0.181 | −3.864 * | 0.000 | −0.14 | −3.07 * | 0.00 | 0.03 | 1.23 | 0.22 |
| Grade | −0.052 | −1.116 | 0.265 | −0.04 | −0.77 | 0.44 | −0.06 | −2.34 * | 0.02 |
| Major in Social Work |  |  |  | −0.11 | −2.36 * | 0.02 | −0.02 | −0.65 | 0.51 |
| Gay contact |  |  |  | −0.16 | −3.46 * | 0.00 | −0.01 | −0.50 | 0.61 |
| Adoption contact |  |  |  | 0.05 | 1.12 | 0.27 | 0.00 | −0.03 | 0.97 |
| Same-sex parenting |  |  |  |  |  |  | 0.86 | 31.04 * | 0.00 |
| $\Delta R^2$ |  | 0.088 |  |  | 0.044 |  |  | 0.603 |  |
| $F_{(\Delta R2)}/p$ |  | 13.78 * | 0.000 |  | 7.16 * | 0.000 |  | 963.48 * | 0.000 |
| $R^2$ |  | 0.088 |  |  | 0.132 |  |  | 0.735 |  |
| $F_{(R2)}/p$ |  | 13.78 * | 0.000 |  | 10.77 * | 0.00 |  | 167.82 * | 0.00 |

* $p < 0.05$.

**Table 9.** Hieratical regression to Idea Recognition by three blocks of predictors.

|  | Block 1: Demographic | | | Block 2: Experience | | | Block 3: Same-Sex Parenting | | |
|---|---|---|---|---|---|---|---|---|---|
|  | Beta | $t$ | $p$ | Beta | $t$ | $p$ | Beta | $t$ | $p$ |
| Biological sex | −0.204 | −4.355 * | 0.000 | −0.18 | −3.92 * | 0.00 | 0.00 | 0.03 | 0.98 |
| Sexual orientation | −0.181 | −3.864 * | 0.000 | −0.14 | −3.07 * | 0.00 | 0.03 | 1.23 | 0.22 |
| Grade | −0.052 | −1.116 | 0.265 | −0.04 | −0.77 | 0.44 | −0.06 | −2.34 * | 0.02 |
| Major in Social Work |  |  |  | −0.11 | −2.36 * | 0.02 | −0.02 | −0.65 | 0.51 |
| Gay contact |  |  |  | −0.16 | −3.46 * | 0.00 | −0.01 | −0.50 | 0.61 |
| Adoption contact |  |  |  | 0.05 | 1.12 | 0.27 | 0.00 | −0.03 | 0.97 |
| Same-sex parenting |  |  |  |  |  |  | 0.86 | 31.04 * | 0.00 |
| $\Delta R^2$ |  | 0.055 |  |  | 0.036 |  |  | 0.397 |  |
| $F_{(\Delta R2)}/p$ |  | 8.26 * | /0.000 |  | 5.61 * | /0.001 |  | 302.39 * | /0.000 |
| $R^2$ |  | 0.055 |  |  | 0.091 |  |  | 0.469 |  |
| $F_{(R2)}/p$ |  | 8.26 * | /0.000 |  | 10.77 * | /0.000 |  | 167.82 * | /0.000 |

* $p < 0.05$.

**Table 10.** Hieratical regression to Action Support by three blocks of predictors.

|  | Block 1: Demographic | | | Block 2: Experience | | | Block 3: Same-Sex Parenting | | |
|---|---|---|---|---|---|---|---|---|---|
|  | Beta | $t$ | $p$ | Beta | $t$ | $p$ | Beta | $t$ | $p$ |
| Biological sex | 0.125 | 2.66 * | 0.008 | 0.11 | 2.27 * | 0.02 | 0.00 | −0.02 | 0.99 |
| Sexual orientation | 0.215 | 4.57 * | 0.000 | 0.19 | 4.03 * | 0.00 | 0.09 | 2.07 * | 0.04 |
| Grade | 0.062 | 1.33 | 0.183 | 0.05 | 1.12 | 0.27 | 0.06 | 1.56 | 0.12 |
| Major in Social Work |  |  |  | 0.16 | 3.42 * | 0.00 | 0.10 | 2.54 * | 0.01 |
| Gay contact |  |  |  | 0.10 | 2.14 * | 0.03 | 0.01 | 0.34 | 0.74 |
| Adoption contact |  |  |  | −0.08 | −1.67 | 0.10 | −0.05 | −1.16 | 0.25 |
| same-sex parenting |  |  |  |  |  |  | −0.50 | −11.41 * | 0.00 |
| $\Delta R^2$ |  | 0.075 |  |  | 0.044 |  |  | 0.207 |  |
| $F_{(\Delta R2)}/p$ |  | 11.49 * | 0.000 |  | 6.98 * | 0.001 |  | 130.16 * | 0.000 |
| $R^2$ |  | 0.075 |  |  | 0.118 |  |  | 0.326 |  |
| $F_{(R2)}/p$ |  | 11.49 * | 0.000 |  | 9.48 * | 0.000 |  | 29.19 * | 0.000 |

* $p < 0.05$.

Figure 1 shows the summary of Tables 8–10. The three same-sex dimensions are predicted by Block 1: demographic variables; Block 2: experiences; and Block 3: same-

sex parenting. The three dimensions of same-sex adoption: "Action Support", "Idea Recognition", and "Worry/Against" are predicted by the independent variables of Block 1, 2, and 3, respectively, and 32.6% (Table 10), 46.9% (Table 9), and 73.5% (Table 8) of the variation are explained, respectively.

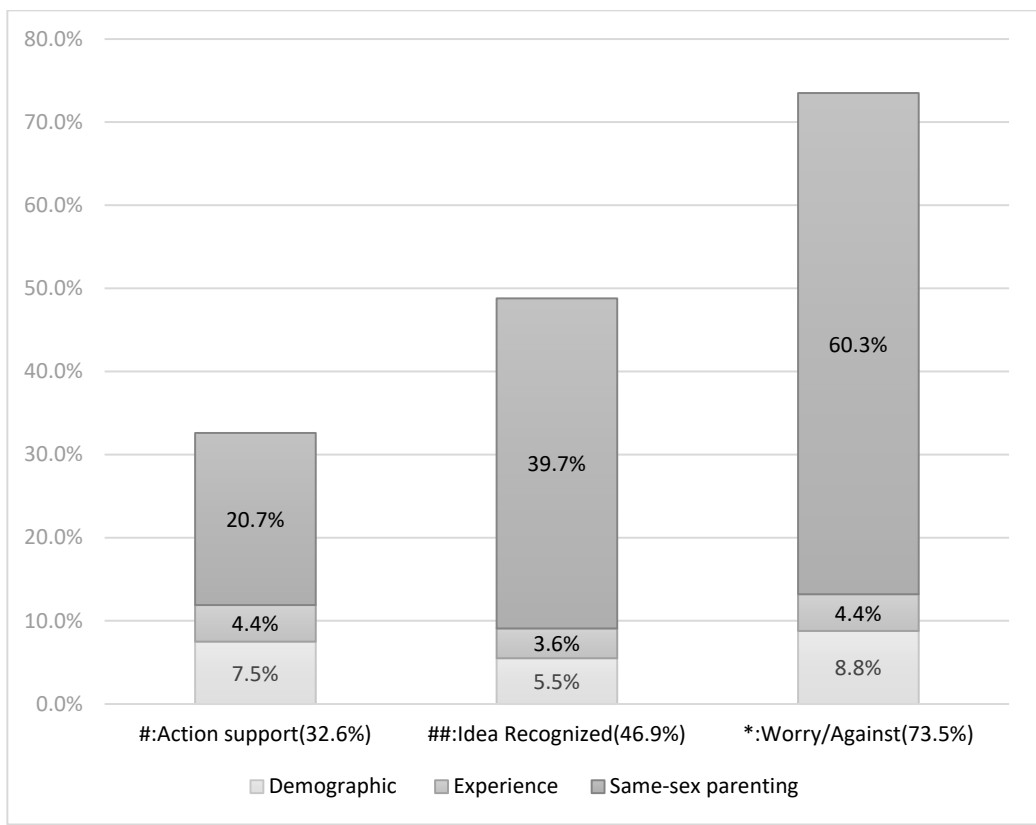

**Figure 1.** The explanation percentage of three dimensions of same-sex adoption. #: refers to Table 10; ##: refers to Table 9; *: refers to Table 8.

Demographic variables, namely, biological sex, sexual orientation, and grade, are put into the regression predictive model as Block 1, and explain 7.5%, 5.5%, and 8.8% of variance, respectively, for the three same-sex adoption dimensions: "Action Support", "Idea Recognition", and "Worry/Against", which are significant and important.

The three independent variables of experience, namely, gay-contact, adoption contact, and professional department contact, are put into the regression model as Block 2, and increased the predictive power of the three same-sex adoption dimensions by 4.4%, 3.6%, and 4.4% of variance, respectively. That are as important as Block 1. More importantly, Block 3, same-sex parenting, increase 20.7%, 39.7%, and 60.3% of the predictive power in the three dimensions of same-sex adoption. These are much larger than Blocks 1 and 2.

Upon examining the dependent variables, it was discovered that the "Idea Recognition" aspect of cognition towards same-sex adoption can be predicted up to 46.9%. In contrast, the "Worry/Against" component of affect reaches a peak of 73.5%, and the "Action Support" aspect of behavior is the lowest at 32.6%. It is not surprising to see such high emotions surrounding the issue of homosexuality. This indirectly confirms Breckler's (1984) theory on attitudes that affect, cognition, and behavior were correlated yet distinct.

The perception of same-sex parenting greatly influences attitudes towards same-sex adoption. Individuals' beliefs about the ability of same-sex couples to effectively raise children significantly impact their stance on whether same-sex couples should be permitted to adopt. Although other demographic variables, such as biological sex, sexual orientation, grade, and exposure to gay individuals or adoption, can also shape attitudes towards gay adoption, the most influential factor is the perception of same-sex couples' parental

capabilities. Despite numerous studies on this subject, the results are still inconclusive. Further research is required to fully comprehend the intricate relationship between attitudes towards same-sex parenting and adoption.

## 4. Discussion

### 4.1. Taiwanese University Students Have Positive Attitude to Same-Sex Marriage Adoption

The research reveals that Taiwanese university students hold a positive attitude towards same-sex marriage adoption. Similar to Dotti Sani and Quaranta's (2020) study on attitudes towards same-sex marriage adoption in 22 European countries, it suggests that individuals who are young and well-educated in countries with more progressive legislation on same-sex relationships tend to have a more favorable view of same-sex marriage and adoption. On a Likert scale ranging from one to five, our study found that the average positive attitude score of Taiwanese university students towards same-sex marriage adoption was higher than three points. In particular, the students showed a more positive attitude towards the "Idea Recognition" aspect than "Action Support". Despite the fact that Taiwan only permitted stepchild adoption after passing the same-sex marriage law in 2019, our research indicates that young people in Taiwan hold a positive attitude towards same-sex marriage adoption.

### 4.2. Female University Students Have a More Positive Attitude to Same-Sex Marriage Adoption versus Male University Students

Men, in particular, have been reported to hold the strongest homophobic or anti-gay attitudes (Costa et al. 2014a; Kemper and Reynaga 2015; Mirabito 2014). Men are generally less likely to accept sexual minorities and equal adoption rights for same-sex couples (Bettinsoli et al. 2020; Webb et al. 2017). Gender factors used to be mentioned in the same-sex marriage research; it usually indicated that men, in particular, have been reported to hold the strongest homophobic or anti-gay attitudes (Costa et al. 2014a; Kemper and Reynaga 2015; Mirabito 2014). Compared to woman, man generally support traditional gender roles. Why did men not meet the same degree as women? Korpi et al. (2013) indicated that the patriarchal system makes it easier for men to identify with men's gender roles by giving them more power and status. That is why men always lag behind women in the evolution of gender issues.

### 4.3. Sexual Orientation and Homosexuality Contact Experience Are Related with Same-Sex Marriage Adoption

Our research findings indicate that sexual orientation and homosexuality contact experience are closely related to attitudes towards same-sex adoption. Specifically, heterosexual individuals tend to show more worry and opposition towards same-sex marriage adoption and have less action support and idea recognition. These attitudes may stem from a lack of understanding and perpetuated stereotypes. Montero (2014) similarly notes that while attitudes towards homosexuality have become more positive, people still question the ability of same-sex couples to foster or adopt children. Heterosexual individuals often view sexual orientation as a fixed genetic trait. Our research suggests that having more experiences with homosexuality can help reduce stereotypes and shift attitudes towards a more positive direction.

Misunderstanding is a common cause of stereotypes. Our research suggests that increasing one's experience with homosexuality can improve understanding and decrease stereotypes. Previous studies have shown that university students who have had more contact with homosexuality have a more positive attitude towards same-sex marriage adoption, with higher levels of Action Support and Idea Recognition compared to those with little or no contact. Our own research confirms this finding, with homosexuality and adopter contact experiences being significant predictors of attitudes towards same-sex marriage adoption, even after controlling for demographic variables.

*4.4. Social Work Students Show Less Worry/Against on Same-Sex Marriage Adoption*

In Taiwan, social workers and the judiciary are the most likely professions to be involved with adoption. According to our research, social work students have the lowest level of opposition towards same-sex marriage adoption; although there is no significant difference in their level of "Idea Recognition" compared to other groups, they do show a significant difference in their level of "Action Support". This finding is encouraging as it highlights the importance of being receptive to multiple cultures in the field of social work.

*4.5. University Students of Different Religions Show Significant Differences in Idea Recognition*

Religion is one of the most predictable factors of same-sex marriage adoption attitude (Scherman et al. 2020). Our research has discovered that Christians have lower 'Idea Recognition' than others. Taiwan has approved a bill legalizing same-sex marriage; during the process, a group led by Christians, "Family Guardian Coalition", was opposed to same-sex marriage; perhaps this could show why Christian have lower 'Idea Recognition' than others, but demonstrate no significant differences in Action Support and Worry/Against. Dose this also means that in Taiwan, people respect others? Taiwanese is an increasingly multicultural society, and the "Family Guardian Coalition" cannot represent all Christians in Taiwan.

**5. Conclusions**

In May 2019, Taiwan became the first country in Asia to legalize same-sex marriage through the Enforcement Act of the Judicial Yuan Interpretation, No. 78. As per Article, 20, same-sex couples are only allowed to adopt stepchildren. However, this has proven to be a disappointment for many same-sex couples who hope to have children. Efforts are being made to amend the law. The restriction on same-sex adoption has had negative effects on many same-sex couples and adoption agencies. Adoption agencies and social workers are disappointed that same-sex marriage adoption did not become a reality.

Taiwanese university students have a positive attitude towards same-sex marriage adoption. Female students have more positive attitudes than male students, and social work students are the most accepting. Sexual orientation and homosexuality contact experiences also influence attitudes towards same-sex marriage adoption. Heterosexual students are generally more worried and against same-sex marriage adoption, while those with more homosexuality contact experience have more positive attitudes towards it. Religion is also a factor, with Christian students having lower idea recognition. Our study highlights the importance of open-mindedness and understanding towards diverse groups for creating positive attitudes. We believe that open-mindedness and understanding are crucial for breaking stereotypes. We encourage LGBTQ individuals to be open about their identity and for adults to recognize the necessity of a multicultural society in today's world.

The three factors of attitudes towards same-sex adoption in this study, "Worry/Against", "Idea Recognition", and "Action Support", are mutually reinforcing with the ABC model. Moreover, the three factors in this study are interrelated; however, as with the ABC model, it remains to be investigated whether affect influences behavior, which then impacts cognition (A-B-C), which would comply with Solomon's Experiential Hierarchy, or the Low-Involvement Hierarchy; cognition influences behavior, which then impacts affect (C-B-A), and other sequential combinations (Solomon 2019).

In addition, our research found that the concept of same-sex parenting is the primary predictor of same-sex marriage adoption. However, more evidence is needed in Taiwanese culture to demonstrate that children in same-sex marriages are well-cared for. We suggest that same-sex couples who have already raised children should speak out about their positive experiences, which would be an Experiential Hierarchy (A-B-C) way to make the change. Moreover, researchers and professionals should also share their findings through news or research publications to help alleviate public concerns and correct any misunderstandings in a Low-Involvement Hierarchy (C-B-A) way.

**Funding:** This research received no external funding.

**Informed Consent Statement:** Informed consent was obtained from all subjects involved in the study.

**Data Availability Statement:** The data presented in this study are partial available from the corresponding author upon request.

**Conflicts of Interest:** All authors declare that they have no conflict of interest.

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
