# Peer review of "University Students Attitudes toward Same-Sex Marriage Adoption in Taiwan"

_socsci, doi:10.3390/socsci12040201_

Round 1

Reviewer 1 Report

Overall, excellent and an important study given newly enacted law. Two recommendations:

In section 4.2 - I would give an additional sentence on either:

1. Why women supported more or

2. why men did not meet the same degree as women.

In conclusion - I would re-write the first sentence and break it down further. The English was not really clear and it's such an important topic legally and socially, should be clearer.

Author Response

Thank you for your positive review and helpful suggestions.

  1. We have added a sentence in section 4.2 to explain the reason.

2.The conclusion have been clarified as your suggestion.

We appreciate your feedback and look forward to implementing these changes.

Reviewer 2 Report

This is an interesting topic area. However, it needs major development for a journal audience. Perhaps ask yourselves whether there is a significant finding for an academic paper. Then try to present the data so that the reader can see the supporting evidence. 

It seems that the main finding is that participants who are gay or bisexual are more in favour of same sex adoption than others, This is not surprising.

Also that social workers are supportive of same sex adoption. This is perhaps more interesting as they are gatekeepers to the process.

The whole piece needs careful editing of the English and also many of the subheadings need developing e.g. 1.3.4 others is not a suitable heading. Also much clearer labels of the variables are needed in the tables e.g  'worry about being repelled' as at present they do nnot make sense for the reader.

There are too many tables and these are not referred to or explained. I would suggest having fewer tables and that these need to be explained in the text so that the reader can see why you have included tham.

Author Response

Thank you for your helpful feedback. We have made significant changes to the manuscript, including adding a more substantial discussion of our main findings and providing clearer explanations of the variables in the tables. We have also carefully edited the English throughout the manuscript, and we have revised the subheadings to better reflect the content.

We acknowledge that the finding on participants who are gay or bisexual being more in favor of same-sex adoption is not surprising. Although not surprising, the finding that individuals from the LGBTQ+ community or those who have contact with them are more supportive of same-sex adoption is significant and underscores the foundation of a multicultural society. Hence, it has been included in the results section.

Regarding the number of tables, we believe that each table is essential in supporting our findings, and we have included them to provide readers with a more comprehensive understanding of our results. We have provided detailed explanations for each table in the text to help readers understand why we have included them.

Once again, thank you for your comments, and we hope that you find our revised manuscript to be more suitable for publication in an academic journal.

Reviewer 3 Report

The study as such is sound when it comes to the empirical part and the theme is highly relevant. Nonetheless there are essential parts that need to be improved.

1. There is an obvious lack of theory in the article and this needs to be corrected. What kind och theories are the observations about formation of attitudes rooted in. What are the main applicable theories regarding how attitudes are formed of relevance to this study.

2. The above needs to be well connected to the previous studies that are reported. These are now presented in a quite fragmented form that does not aid the over-all understanding. The article needs a firm presentation of the state of art and this needs to be connected to the theoretical framework.

3. Research questions need to emerge from the above. How does the current study of attitudes in the specific sample provide further knowledge that has some larger scientific relevance. Or is this just another study that confirms what we already know from proviso research?

4. The argument and conclusions that come out of the article needs to contribute with something new or specifically relevant. The authors also need to reflect critically on their results. To what extent is the fact that "gay parenting attitudes are the most critical predictor of gas adoption attitudes" is surprising at all? Could it be any other way?

5. In an academic and scientific article of this kind wikipedia articled cannot be cited as reliable information or knowledge. Only when it comes to academic publications can we be sure that the knowledge or information presented has been produced with teh methods and review that ensures this. 

6. The language is not good enough. Most parts are clear and good in terms of grammar and style but some parts are unfortunately unintelligible. Even though the latter are few, the problem affects certain important parts in a way that makes the whole article suffer. 

The article needs major revision but I  would encourage the authors to continue. The study is sound and most probably interesting and relevant. I look forward to a new version.

Author Response

Thank you for your helpful feedback. We do many revise as follow:

1.Breckler’s theory about attitude was added, that is very helpful.

2.We had try to make the result connected to the theoretical framework.

  1. This study is not about confirming what we already know, but about responding to changes in legislation and the environment, and trying to build a stronger foundation for the vision of multiculturalism.

4.We have tried to say that in other way. Although not surprising, the finding that same-sex parenting is the most critical predictor reveal a fact that we all know but neglect. It is important that we face the fact, and try not overlook as usual.

5.The citation of Wiki was deleted, we apology for that.

6.We have fix the English as possible.

Thanks again for the encouragement. We all try to make the world better through the research article.

Round 2

Reviewer 2 Report

The English is improved but needs some more work. In particular the last 3 paragraphs of the results and the discussion section need very careful revision.

The authors wish to leave in all the tables. I think this paper would be much improved by omitting Tables 4, 5, 6 and 7 and describing them in the text. 

Otherwise I am happy with the amendments.

Author Response

1.The English on results and discussion has been revised carefully. 

2.We found that Table 4,5,6 and 7 are necessary for emperical thinking readers, and decide to keep them.

3.We change Table 11 to figure 1, to increase clarity

Thank you for all your help.

Reviewer 3 Report

I think the authors have done an excellent work with the revision and the article now meets the standards. It is ina interesting and relevant study. The theoretical part is not sufficiently presented but the authors would still need to clarify how this use of the ABC model is reflection in the measures used in the study. This should not be difficult. And concerning the results: could one assume that that working simultaneously and together with the ABC dimensions makes a stronger case for attitudinal change? 

Author Response

Thank you for your affirmation.  We added some words about the instrument formation, which fit the ABC model.  And concerning the Attitude Change, We believe that both affect and cognition intervention would help.

Thank you again for all your help